# Telecoupling Research: The First Five Years

**Kelly E. Kapsar** [1,*,†] **, Ciara L. Hovis** [1,*,†] **, Ramon Felipe Bicudo da Silva** [2] **,
Erin K. Buchholtz** [3] **, Andrew K. Carlson** [1,4] **, Yue Dou** [1]**, Yueyue Du** [5]**, Paul R. Furumo** [6]**,
Yingjie Li** [1,7] **, Aurora Torres** [8,9]**, Di Yang** [10]**, Ho Yi Wan** [11]**, Julie G. Zaehringer** [12]
**and Jianguo Liu** [1]

[1]  Center for Systems Integration and Sustainability, Michigan State University, East Lansing, MI 48823, USA;
    carls422@msu.edu (A.K.C.); yuedou.whu@gmail.com (Y.D.); liyj@msu.edu (Y.L.); liuji@msu.edu (J.L.)
[2]  Center for Environmental Studies and Research, State University of Campinas, Campinas,
    SP 13083-870, Brazil; ramonbicudo@gmail.com
[3]  Ecology & Evolutionary Biology Program, Texas A&M University, College Station, TX 77843, USA;
    ebuchholtz@tamu.edu
[4]  Program in Ecology, Evolutionary Biology, and Behavior, Michigan State University,
    East Lansing, MI 48824, USA
[5]  College of Urban and Environmental Sciences, Peking University, Beijing 100871, China;
    duyueyue91@foxmail.com
[6]  Department of Environmental Science, University of Puerto Rico-Rio Piedras, San Juan, PR 00931, USA;
    pfurumo@gmail.com
[7]  Environmental Science and Policy Program, Michigan State University, East Lansing, MI 48824, USA
[8]  German Centre for Integrative Biodiversity Research (iDiv) Halle-Jena-Leipzig, 04103 Leipzig, Germany;
    aurora.torres@idiv.de
[9]  Institute of Biology, Martin Luther University Halle-Wittenberg, 06108 Halle (Saale), Germany
[10]  Department of Geography, University of Florida, Gainesville, FL 32611, USA; yangdi1031@ufl.edu
[11]  School of Public and Community Health Sciences, University of Montana, Missoula, MT 59812, USA;
    hoyiwan@gmail.com
[12]  Centre for Development and Environment, University of Bern, Mittelstr. 43, 3012 Bern, Switzerland;
    julie.zaehringer@cde.unibe.ch
*   Correspondence: kelly.kapsar@gmail.com (K.E.K.); ciarahovis@gmail.com (C.L.H.)
†   Co-first authors.

**Abstract:** In an increasingly interconnected world, human–environment interactions involving flows of people, organisms, goods, information, and energy are expanding in magnitude and extent, often over long distances. As a universal paradigm for examining these interactions, the telecoupling framework (published in 2013) has been broadly implemented across the world by researchers from diverse disciplines. We conducted a systematic review of the first five years of telecoupling research to evaluate the state of telecoupling science and identify strengths, areas to be improved, and promising avenues for future study. We identified 89 studies using any derivation of the term telecoupling. These works emphasize trade flows, information transfer, and species dispersal at international, national, and regional scales involving one or a few countries, with China, Brazil, and the United States being the most frequently studied countries. Our review showed a rising trend in publications and citations on telecoupling, with 63% of identified telecoupling studies using the framework's specific language (e.g., "flows", "agents"). This result suggests that future telecoupling studies could apply the standardized telecoupling language and terminology to better coordinate, synthesize, and operationalize interdisciplinary research. Compelling topics for future research include operationalization of the telecoupling framework, commonalities among telecouplings, telecoupling mechanisms and causality, and telecoupled systems governance. Overall, the first five years of telecoupling research have improved our understanding of human–environment interactions, laying a promising foundation for future social–ecological research in a telecoupled world.

**Keywords:** telecoupling; coupled human and natural systems; social–ecological systems; human–environment interactions; systems integration; globalization; interdisciplinary research

## 1. Introduction

Our world is becoming increasingly interconnected through flows of organisms, people, energy, matter, financial assets, information, and technology [1–5]. Furthermore, human–environment interactions in a single location (e.g., agriculture, forestry, ecotourism) can have substantial influence over other distant places through processes such as global trade, migration, and information transfer [6,7]. These interactions have profound implications for sustainability research and management [8]. Because of the complexity and differences in environments and policies across geographic regions and sociopolitical boundaries, understanding and governing these distant interactions is often challenging.

In 2013, Liu et al. [9] introduced an integrated framework of telecoupling to systematically describe socioeconomic and environmental interactions among distant coupled human and natural systems. In the first five years of its existence, over 300 papers have cited Liu et al. [9] (Figure 1). The telecoupling framework has been broadly implemented in science communities across the world, in both terrestrial and aquatic environments, and from the lens of a wide array of disciplinary and interdisciplinary approaches.

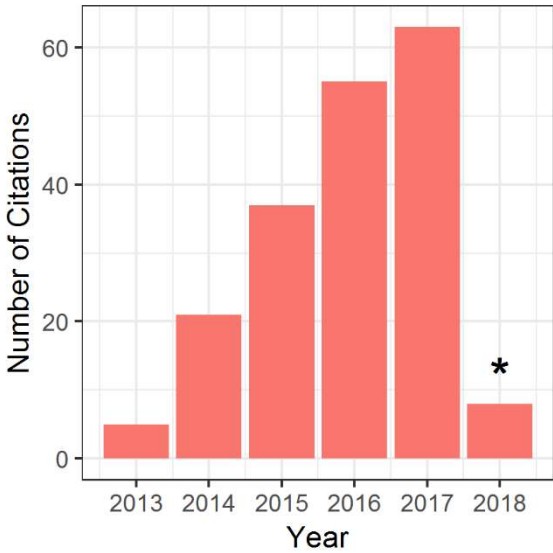

**Figure 1.** Number of citations per year for "Framing Sustainability in a Telecoupled World" [9]. * Data extend through 1 July 2018.

The telecoupling framework builds upon a foundation of disciplinary research, particularly environmental science, climatology, and systems science research. For example, the similar concept of "teleconnection" is frequently used by climatologists to explain the impacts of certain climatological processes on distant locations [10]. Additionally, the term 'teleconnection' emerged as a concept for understanding processes behind land system change in the context of human and natural influences [11–13]. However, these approaches lack the structure and conceptual breadth needed to encompass the interrelated social and ecological aspects of complex systems. The telecoupling framework provides a universal and hierarchical model with specific language to better identify and understand distant interactions in and among coupled human and natural systems (CHANS) by distinguishing "sending", "receiving", and "spillover" systems and their environmental and socioeconomic components (i.e., "causes", "effects", and "agents") within user-defined boundaries

(Figure 2). The connections between systems are defined as "flows", which can be either tangible (e.g., organisms, people, materials) or intangible (e.g., capital, knowledge, technology).

Another example of the semantic integration of the telecoupling framework is the concept of a spillover system (Figure 2c). Many words have been used in different disciplines to describe the unintended consequences of particular actions (e.g., displacement, off-site impact, spatial externality, leakage). However, the concept of a spillover system integrates unintended consequences into the telecoupling process such that the spillover system is recognized as a part of a much larger system. This contextualization lays a foundation for systematically and consistently predicting the potential impacts of different policy actions and promoting the sustainable development of complex systems.

In the course of the five years since its introduction, the telecoupling framework has rapidly been adopted by many scientific disciplines, including earth science, ecology, economics, environmental science, fisheries, and political science [14–19]. Its applications have informed a wide range of important issues, such as sustainability management, conservation, ecosystem services, international trade, food security, energy, and tourism [3,20–25]. Given the rapid adoption and growing influence of telecoupling, it is important to review the current progress, summarize findings from telecoupling research, and discuss future research directions. In this paper, we examine how the concept of telecoupling and the telecoupling framework have been used in research, as well as how they have grown and evolved during their first five years. Our objective is to gain a thorough understanding of the state of telecoupling research and to identify strengths, areas to be improved, and promising avenues for new research. To do so, we reviewed all telecoupling papers published in scientific peer-reviewed journals since the inception of the telecoupling framework and recorded an array of metrics, including number of citations, study areas, methods used, and study topic(s). The results of this review paint a clear picture of the field of telecoupling research and help to identify research areas in which telecoupling has a strong presence, as well as those in which there is potential for expansion.

## 2. Review Criteria

Our review was guided by a rigorous protocol for the systematic review of literature, the PRISMA (preferred reporting items for systematic reviews and meta-analyses) framework [26]. A literature search was conducted on the Web of Science (WOS) database (accessed on 1 July 2018). Per our protocol (see Supplementary Material), we identified potential articles using a topic search for all forms of the word "telecoupling" published between 1 January 2013 and 1 July 2018. Papers too recent to appear in the WOS database (i.e., published around 1 July 2018) were identified by co-authors and also included in our review. Gray literature, such as nonformally published or peer-reviewed theses, dissertations, reports, and working papers, were excluded from this analysis.

Our literature search yielded 89 publications fitting our search criteria, with 15 of those added after the initial WOS search. All papers were thoroughly reviewed and classified into one of three categories (Figure 2). "Phenomenon" papers were those that mentioned the word telecoupling, but outside of the context of the research that was being conducted (e.g., telecoupling was a keyword or appeared in the abstract but was not mentioned any other time). "Concept" papers used the word telecoupling at least once in the context of the research being conducted. "Framework" papers applied the telecoupling framework by labelling at least one component of the study system(s) according to the telecoupling framework (e.g., identified sending and receiving systems; full details in the Supplementary Material). We then coded each paper and extracted information on the spatial and temporal scale(s) and extent(s) examined, names and number of countries studied, and telecoupled flow type.

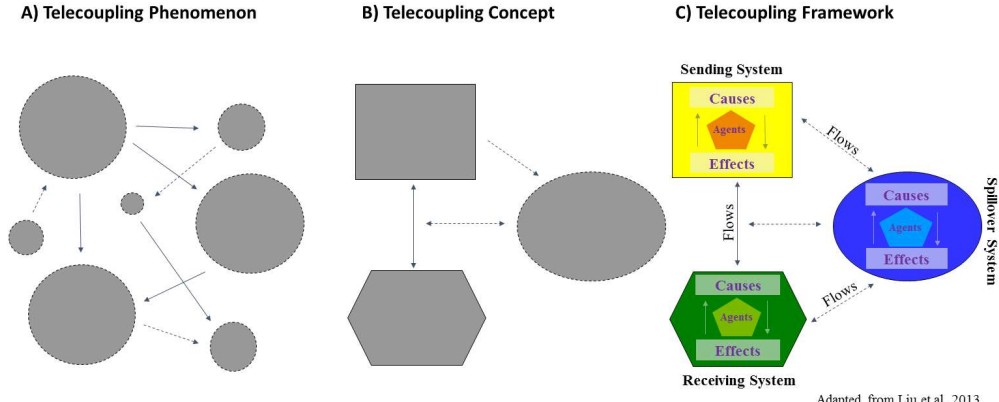

**Figure 2.** Visual depictions of the ways in which telecoupling is used in the literature. (**A**) Phenomenon: The acknowledgement of connections between distant places. Boundaries and roles of places not clearly defined. (**B**) Concept: Studied components or systems are characterized as affected by a telecoupled process, with some boundaries defined. However, the specific terminology of the telecoupling framework is not used. (**C**) Framework: Explicit usage of the specific language of the telecoupling framework with boundaries and system roles clearly defined. Circles and gray coloration are used to represent undifferentiated 'systems', while color and shape distinctions are abstract indications of different levels of adherence to the telecoupling framework. Dashed outlines represent unspecified system boundaries while solid lines surround clearly defined systems.

## 3. Overview of Five Years of Telecoupling Research

The amount of telecoupling research shows a clear, increasing trend over time (Figure 3B). Papers mentioning the word telecoupling in their title, abstract, or keywords have been published in 47 different peer-reviewed journals and books in the past five years. *Ecology & Society* is the largest venue for telecoupling research, having published 20 telecoupling papers since 2013 (Figure 3A).

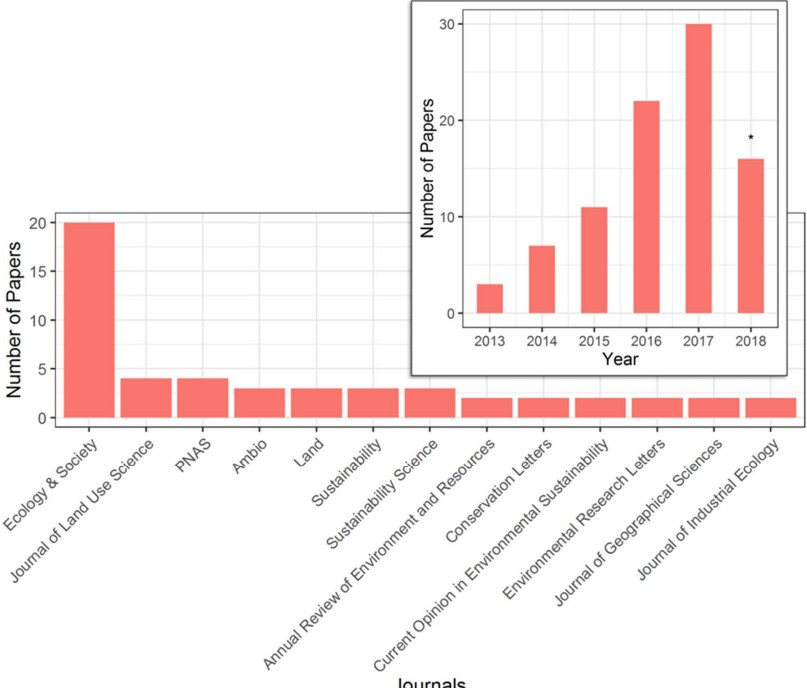

**Figure 3.** (**A**) Frequency of telecoupling papers published in academic journals (excludes journals that have only published one telecoupling paper). (**B**) Number of papers published that mentioned "telecoupling" by year. * Data in 2018 extend through 1 July 2018.

Of the 89 papers that were reviewed, 11 were classified as Phenomenon, 24 as Concept, and 54 as Framework (Figure 4). Telecoupling flow type, disciplinary focus, and country information were analyzed for the 54 Framework papers.

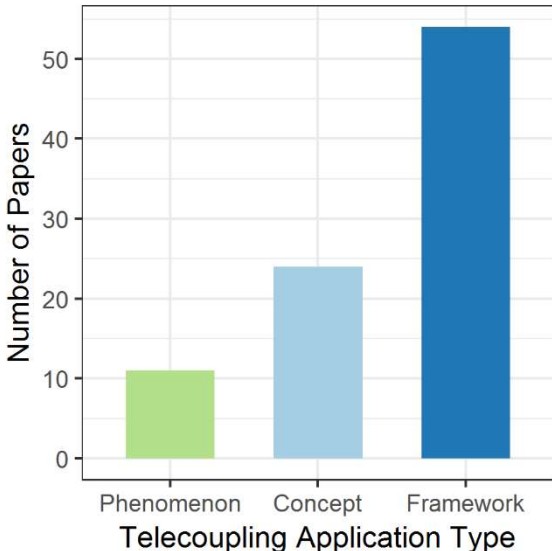

**Figure 4.** Frequency of each telecoupling use category in the literature review (*n* = 89).

We identified several telecoupling flow types from the Framework papers, including trade, knowledge transfer, species dispersal, tourism, water transfer, human migration, waste transfer, biophysical, technology transfer, investment, animal migration, and ecosystem services flow. Seventy-four percent of the 54 papers addressed trade, 33% addressed knowledge transfer, 17% addressed species dispersal, followed by tourism, water transfer, and other flows, which each accounted for less than 10% of papers (Figure 5a).

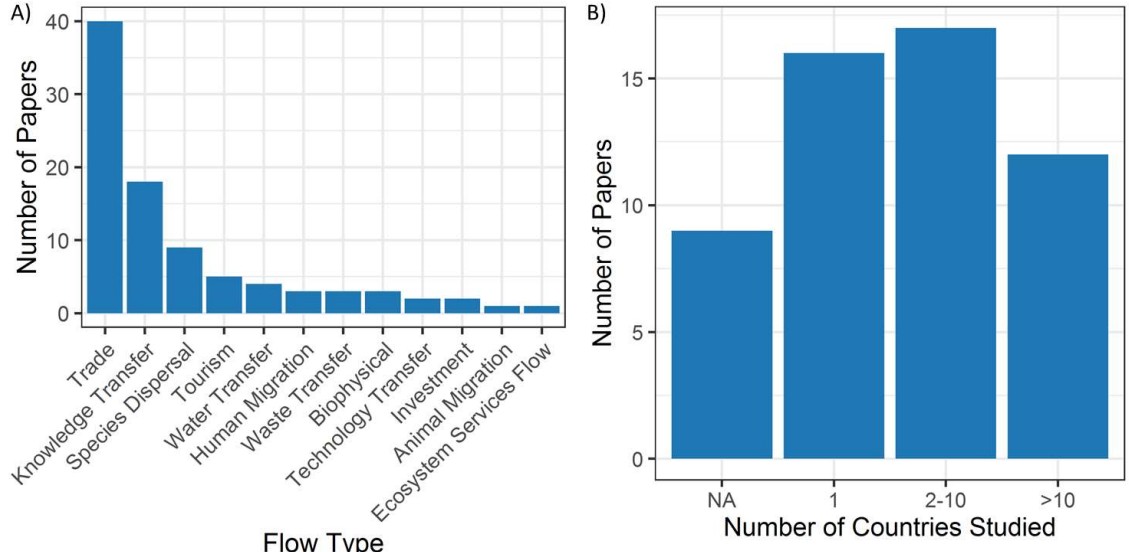

**Figure 5.** (**A**) Frequency of telecoupling flow types analyzed in framework papers. (**B**) Frequency of the number of countries studied in framework papers. Papers with NA indicate studies that apply the framework, but not in the context of specific countries (e.g., publications examining the general utility of the telecoupling framework for better understanding biodiversity conservation or trade).

The geographic scale of analysis describes the level at which data were analyzed (local, regional, national, international). Of the Framework publications, 14 papers focused on telecoupling at an

international scale, 13 focused on a regional or national scale, and 6 focused on a local scale. Twelve papers investigated multiple geographic scales across local, regional, national, and international boundaries. The number of countries analyzed ranged widely, with a minimum of one and a maximum of 172 countries examined in a single study. However, most Framework papers (61%) focused on one or a few countries, with just 12 papers analyzing more than 10 countries (Figure 5b).

Excluding analyses with >10 countries, studies applying the telecoupling framework (i.e., Framework papers) examined 34 countries on 6 continents (Figure 6; *n* = 33 papers). China was the most frequently studied country, included in 19 papers. Brazil, Laos, and the United States appeared in at least 4 papers each (Figure 6).

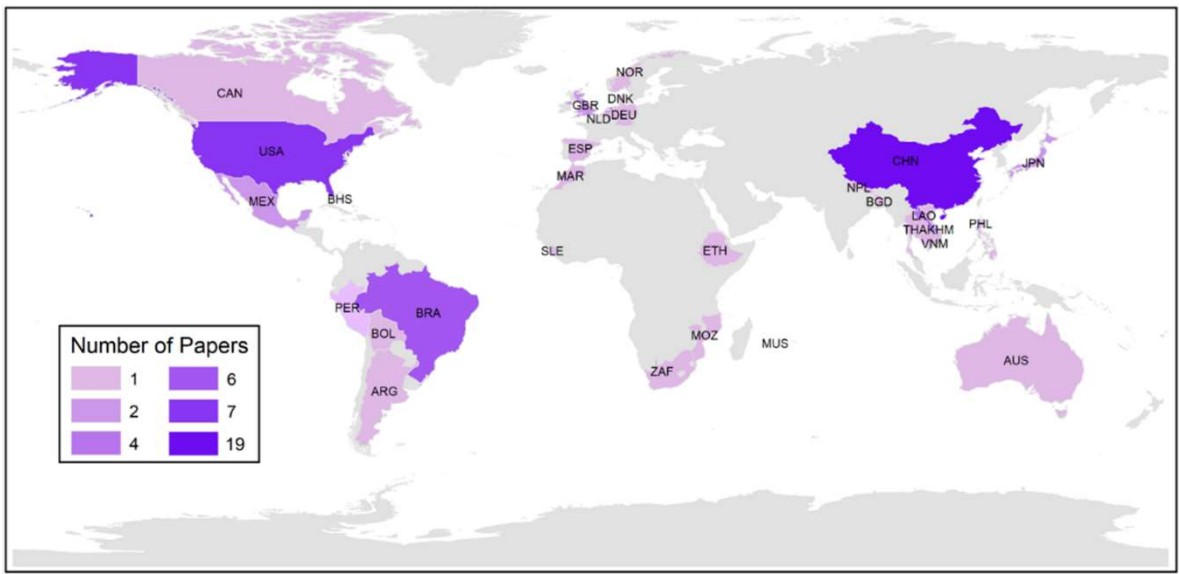

**Figure 6.** Map of the number of studies conducted in each country using the telecoupling framework (excludes analyses with >10 countries; *n* = 33 publications). Country codes follow the International Organization for Standardization 3166-1 Alpha-3 standard.

## 4. Emergent Themes from the Telecoupling Literature

This review demonstrates that the telecoupling framework has rapidly been adopted in the five years since its inception. It has been applied across the gamut of sustainability science to better understand international trade [3], land use/land cover change [27,28], tourism [29], water transfer [30,31], species invasion [32], migration [18], and urbanization [33], among many other topics.

However, despite its growing recognition, our results show that 37% of telecoupling papers do not use the specific language of the framework (e.g., "agents", "sending/receiving systems", "flows"). This lack of a unifying language highlights a missed opportunity for the synthesis of telecoupling research.

Still, the original telecoupling framework paper has informed and motivated a number of notable contributions to a variety of disciplines. These contributions can be broadly categorized into four areas: Novel hypothesis generation, the identification of knowledge gaps, the assessment of feedbacks, trade-offs, and synergies of telecoupled systems, and the identification of spillover systems.

The generation of novel hypotheses that result from telecoupling studies can be attributed to the explicit identification of the social and environmental components of both the sending and receiving system. This leads to the development of novel hypotheses that often challenge conventional thinking. For example, in Sun et al.'s paper, the authors hypothesized that the trade of agricultural goods does not always convey environmental benefits for the importing country [34]. They showed that in the case of the soybean trade between Brazil and China, Chinese farmers began growing more maize than soybeans, which increased their nitrogen fertilizer use substantially, thus increasing potential

environmental impacts. In this case, the inclusion of the Brazil–China telecoupling resulted in a new hypothesis that runs counter to the conventional wisdom which states that importing countries' natural environments benefit from trade.

Similarly, the telecoupling framework is well-suited to the identification of knowledge gaps in CHANS research. Gasparri and Waroux exemplified this in their study, where they assessed policy challenges in the South American soybean and cattle industries [35]. They concluded that these interconnected industries require more actor-centered approaches for both conservation policy and research.

The integration of natural and human elements inherent in the telecoupling framework highlights the feedbacks, tradeoffs, and synergies not apparent when treated separately. For example, Cease et al. used the telecoupling framework and were among the first to acknowledge the intertwined nature of livestock prices, locust outbreaks, soil nitrogen, and local livelihoods in Northeast China [35,36], thus improving the understanding of feedbacks in a telecoupled system.

Finally, the identification of spillover systems and the impacts of telecouplings upon them is a major strength of the telecoupling framework. Chignell and Laituri illustrated this by revealing underlying social–hydrological relationships in the context of water development aid in Ethiopia [37]. In this example, the sending system is defined as the international aid sector with rural livelihood systems, urban sanitation services, and producers of hydropower as the receiving systems. Their results indicate that there were unforeseen effects of water development aid on seemingly disparate sectors (i.e., irrigated agriculture and agricultural exports, both defined as spillover systems). In general, these early themes in the telecoupling literature have aided and facilitated a better understanding of sustainability for CHANS in numerous capacities and together create a solid foundation for future telecoupling research directions.

## 5. Future Directions for Telecoupling Research

The dramatic increase in the amount of research conducted suggests that the future of telecoupling will be fruitful. While telecoupling research has already helped investigators to better understand complex CHANS, there is still much work to be done and many areas for expansion and improvement. This section highlights a few examples.

To advance the study of telecoupled system dynamics, further operationalization of the telecoupling framework is warranted. Several techniques and tools are being developed with the purpose of facilitating this progress and improving our understanding of complex interactions between CHANS across distances. One such technique is a telecoupled agent-based model (TeleABM) that simulates telecoupling processes (e.g., international trade, socioeconomic and environmental effects) and can provide a quantitative approach to better understand these complex systems [38]. The model can also be informed with empirical data, which can produce more realistic modeling outcomes. Additionally, a growing collection of software tools and applications, called the Telecoupling Toolbox (http://telecouplingtoolbox.org/), provides researchers and practitioners with multiscale visualization and geoprocessing tools that integrate socioeconomic and environmental analytic methods for the study of telecoupled systems [39]. Its two main products are an ArcGIS Toolbox (Telecoupling Toolbox) and a Telecoupling GeoApp. The latter is a new web-based GIS application that provides researchers and practitioners with a useful platform to explore systems, flows, agents, causes, and effects of telecouplings with a range of simple to complex mapping and geospatial analysis tools (https://telecoupling.msu.edu/geo-app/ [40]).

There are also several key areas where telecoupling research can be expanded. Spillover systems are integral components of almost any telecoupling process. En route from sending to receiving systems, telecoupled flows often generate one or more spillover systems. However, the connections of spillover systems to their telecoupled flows are understudied [41]. For example, while much research has been conducted on the environmental, economic, and social impacts of the global shipping

industry that carries the vast majority of globally traded goods [42–45], little research has connected these 'spillover' effects with the sending and receiving systems ultimately driving shipping demand.

The telecoupling framework has great potential for encouraging researchers to examine the broader context in which their research takes place through cross-disciplinary telecoupling analyses. However, while many researchers seem to acknowledge this context (as seen by the numerous references to telecouplings in the introductions of research papers), there is a need for more interdisciplinary collaborations to analyze all of the components of telecoupling processes from appropriate disciplinary perspectives. Applying the telecoupling framework to better understand complex interrelationships among seemingly unrelated fields could help researchers and practitioners to identify key agents, flows, and feedbacks in these systems. While we acknowledge that this cannot be always accomplished in a single paper, we suggest the idea of a "paper series" in which multiple papers analyze different aspects of a particular telecoupling. Papers in the series can then be integrated together into a more holistic understanding of telecoupled system dynamics. For example, Sino–Brazilian soy trade has been extensively studied under the telecoupling framework (>10 publications), and explicitly linking these findings and understanding their implications could enhance the process of knowledge integration and synthesis for policy-making [3,12,28,34,35,46–50].

Beyond individual telecouplings, there is also a need for research examining the processual commonalities behind telecoupling processes in order to uncover the underlying structure(s) and mechanism(s) of complex, telecoupled systems. For instance, some telecoupling processes are consciously initiated through policies, such as trade deals, while others, such as the illegal wildlife trade, are in part due to an unmet demand for a particular resource. Identifying these shared characteristics of different telecoupling processes can help to illuminate the connections between and among disparate situations and reveal avenues that have been successful for promoting sustainable development. This concept of processual commonalities is very similar to the idea of common pool resources and the common factors that contribute to their sustainable use developed by Nobel Prize winning economist Elinor Ostrom [51]. In terms of methodologies that could also be applied to the analysis of telecoupling processes and their effects on people and the environment, archetype analysis has proved to be very useful in identifying socioecological patterns and recurring processes [52,53].

Furthermore, increased attention should be given to establishing causal effects and causal mechanisms in telecoupled processes [54]. In this Special Issue, Carlson et al. [55] found that less than three percent of published telecoupling papers have applied rigorous (i.e., qualitative–quantitative) causal analysis methods. Improving the analysis of causality in telecoupling research will help to reveal the mechanisms by which telecoupled systems function and processes occur. Much of the telecoupling literature is descriptive. That is, processes are described qualitatively in terms of their causes (e.g., policy), manifestations (e.g., increased market prices/trade), and effects (e.g., land use change). Future work will benefit from identifying how global forces actually exert change at the local level. For instance, how do smallholders make decisions in the face of so many interlinked factors, such as global market prices, that present both new challenges and opportunities? How are livelihoods adjusted or changed, and what are the impacts of these changes on natural systems?

The world's increasing connectivity poses many challenges for sustainable development. Eakin et al. [56] have emphasized the role of governance and institutional change in telecoupled interactions. Nevertheless, the governance of telecoupled systems remains a crucial but widely under-researched field. Political science scholars have recognized this research gap and advocate for in-depth and large sample size comparative empirical studies to identify governance and policy options and institutional systems that can effectively and legitimately govern telecoupled systems [14,57]. Oberlack et al. [58] linked the telecoupling concept with the established concept of polycentric governance [59] and operationalized them through a network of action situation analyses. More emphasis should be put on operationalizing such linkages of place-based analysis of land use changes with process-based analysis of flows and actor networks. Methods such as social network analysis can thus help to identify key actors in telecoupled networks, which might

represent leverage points of change [60]. To identify power differentials between different actors connected to these networks, it is crucial to conduct in-depth investigations of actors' agency and their claims on land. Understanding actors' agency and the networks of flows and institutions they are embedded in is a prerequisite to designing and implementing transformative approaches for sustainable development. These efforts can also lead to a better understanding of the challenges and opportunities for transboundary governance, such as improving the sustainability of commodity supply chains through certification programs and zero deforestation agreements [22,61,62].

Some limitations of the telecoupling framework were identified early on (e.g., rigidity of 'distant' interactions, differences in nature of the connections between faraway and nearby systems). Addressing these limitations necessitated a holistic, multiscale approach. In answer to these limitations, the metacoupling framework was introduced as an umbrella concept for CHANS research in a globalized world. In addition to telecoupling processes, the metacoupling framework provides a shared language to analyze connections within a system as well as between adjacent systems [63]. In other words, the metacoupling framework treats telecouplings as one component of a metacoupled system that connects distant systems along with intracouplings (human–nature interactions within a system) and pericouplings (connecting adjacent systems). Differentiating and integrating these three types of 'sub-systems' within a larger metacoupled system elevates our perspective and furthers our ability to uncover connections not apparent at the single system scale [60,64,65].

## 6. Conclusion

Many of the important contributions of the telecoupling framework are attributable to its novel, systematic, and socioecologically integrative structure. As a paradigm that explicitly synthesizes socioeconomic and environmental information over distances, the telecoupling framework advances monothematic research approaches (i.e., those that only consider human or environmental perspectives at a single place) by providing a social–ecological context for distant interactions and the systems, flows, agents, causes, and effects involved. In so doing, the telecoupling framework facilitates a broader, deeper understanding of complexity than was formerly possible. The telecoupling framework is highly flexible (i.e., useful in diverse fields) and applicable (i.e., conducive for translating science into policy and management strategies). Ultimately, these characteristics of the telecoupling framework allow researchers and managers, among others, to advance the science and practice of sustainability in ways that use information on both local and distant socioeconomic and environmental interactions. This unique ability separates the telecoupling framework from many other research approaches and helps to explain its growing importance across diverse disciplines. The past five years have resulted in a solid foundation of telecoupling research with a future even more promising as new avenues for investigation and application arise.

**Supplementary Materials:** The following are available online at http://www.mdpi.com/2071-1050/11/4/1033/s1, Table S1: PRISMA Framework Checklist, Table S2: Codebook, Table S3: Reviewed Telecoupling Literature.

**Author Contributions:** Conceptualization, J.L.; data curation K.E.K., C.L.H., R.F.B., E.K.B, A.K.C., Y.D. (Yue Dou), Y.D. (Yueyue Du), P.R.F., Y.L., A.T., H.Y.W., and J.G.Z; investigation K.E.K., C.L.H., R.F.B., E.K.B, A.K.C., Y.D. (Yue Dou), Y.D. (Yueyue Du), P.R.F., Y.L., A.T., H.Y.W., and J.G.Z; methodology C.L.H. and K.E.K.; writing—original draft preparation K.E.K., C.L.H., R.F.B., E.K.B, A.K.C., Y.D. (Yue Dou), Y.D. (Yueyue Du), P.R.F., Y.L., A.T., H.Y.W., and J.G.Z.; formal analysis K.E.K. and C.L.H.; writing—review and editing, K.E.K., C.L.H., R.F.B., E.K.B, A.K.C., Y.D. (Yue Dou), Y.D. (Yueyue Du), P.R.F., Y.L., A.T., D.Y., H.Y.W., J.G.Z, and J.L. ; visualization, K.E.K., C.L.H, R.F.B., and Y.L.; supervision, J.L.; project administration, K.E.K. and C.L.H.

**Funding:** J.G.Z. was supported by the Swiss Programme for Research on Global Issues for Development (r4d programme), which is funded by the Swiss National Science Foundation (SNSF) and the Swiss Agency for Development and Cooperation (SDC), under grant number 400440 152167. A.K.C. was supported by the University Distinguished Fellowship and Robert C. Ball and Betty A. Ball Fisheries and Wildlife Fellowship [Michigan State University], the NASA-MSU Professional Enhancement Award, and the United States Department of Agriculture National Institute of Food and Agriculture [grant number MICL04161]. Y.L. was supported by the Environmental Science and Policy Program Fellowship. C. H. and Y. Dou were supported by the National Science Foundation Complex Dynamics of Telecoupled Human and Natural Systems, grant number 1518518. E.K.B. was supported by the NASA-MSU Professional Enhancement Award. K.E.K. was supported by the William W. and Evelyn M.

Taylor Endowed Fellowship for International Engagement in Coupled Humans and Natural Systems. R.F.B.S. was supported by grants from Fundação de Amparo à Pesquisa do Estado de São Paulo (grant numbers 15/25892-7 and 14/50628-9). Y.Du. was supported by the NASA-MSU Professional Enhancement Award and China Scholarship Council. D. Y. was supported by NASA-MSU Professional Enhancement Award. J.L. was supported by the U.S. National Science Foundation, NASA, and Michigan AgBioResearch. . A.T. was supported by the NASA-MSU Professional Enhancement Award.

**Acknowledgments:** We thank the guest editors for inviting us to write this paper and the organizers of the 2018 US-International Association of Landscape Ecology conference for providing a venue for this collaboration to be possible. We thank the reviewers of this publication for their time and insightful comments.

**Conflicts of Interest:** The authors declare no conflict of interest.

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
