# Peer review of "Telecoupling Research: The First Five Years"

_sustainability, doi:10.3390/su11041033_

Round 1
Reviewer 1 Report
The theme of the paper is very interesting. However, methodology and results are very weak. Most analyses in ‘3.Overview of Five Years of Telecoupling Research’ are just descriptive statistics which research portal webs such as SCOPUS, SCIVAL suggest. Explanations about telecopuling literature do have enough grounds. How could you predict future with just some paper? If your paper is in quantitative approach, suggest more elaborated statistical analysis. Otherwise, you should suggest research framework from systemed research review and do contents analysis for qualitative approach.
Author Response
Thank you very much for your review. Please see attached word document for our comments.

Reviewer 2 Report
In this manuscript, the authors present the results of a systematic review of research on telecoupling and telecoupling systems. They looked at 89 carefully selected studies and classified the papers into three useful main categories: phenomenon papers, concept papers, and frameworks papers with the bulk of studies belonging to the latter category. Beyond some descriptive statistics, addressing issues like the flow types looked at and the geographical scale focussed on, they identify and discuss a set of emerging themes and (prevalent) research gaps, and subsequently derive a number of avenues for future research on telecoupled systems in general, and on specific dimensions of these systems including governance and spillovers. To the knowledge of the reviewer, the review nicely and comprehensively covers the main literature on telecoupled systems, provides useful categories sorting the wide array of papers, and identifies interesting and highly relevant patterns and research gaps. The suggestions for future research on telecoupling make sense and capture indeed the main aspects and dimensions. The methodology used appears to be appropriated and the figures presented nicely illustrate the main features identified. This manuscript seems to be part of a Special Issue on the topic of telecoupling so that it can be assumed that some of those issues raised are discussed in more detail in other contributions to this Special Issue. In my opinion, apart from some very few and minor editorial mistakes which the authors should take care of before the article is printed, there are no reasons preventing publication in its current state and form in Sustainability. Line 192: “benefits for the importing” Lines 196f.: “that importing countries’ natural” Lines 292: “of land-use changes” SM: Some minor mistakes and formatting imperfections.
Author Response

(The authors gave the same response as above.)

Reviewer 3 Report
It would be interesting if some exemples have been taken from all de bibliographie read in order to illustrate de interactions referred.
Author Response

(The authors gave the same response as above.)

Round 2
Reviewer 1 Report
The paper is well revised.